# Classroom Dust-Bound Polycyclic Aromatic Hydrocarbons in Jeddah Primary Schools, Saudi Arabia: Level, Characteristics and Health Risk Assessment

**DOI:** 10.3390/ijerph17082779

**Published:** 2020-04-17

**Authors:** Mansour A. Alghamdi, Salwa K. Hassan, Noura A. Alzahrani, Marwan Y. Al Sharif, Mamdouh I. Khoder

**Affiliations:** 1Department of Environmental Sciences, Faculty of Meteorology, Environment and Arid Land Agriculture, King Abdulaziz University, P.O. Box 80208, Jeddah 21589, Saudi Arabia; malshareef44@yahoo.com (M.Y.A.S.); mkhader@kau.edu.sa (M.I.K.); 2Air Pollution Department, National Research Centre, El Behooth Str., Dokki, Giza 12622, Egypt; salwakamal1999@gmail.com; 3Office of Education/South Jeddah (Girls), Department of Primary Grades, Ministry of Education, Jeddah 23524, Saudi Arabia; naz1407@hotmail.com

**Keywords:** PAHs, level, characteristics, health risk, classrooms air conditioner filter dust, primary schools, Jeddah

## Abstract

Data concerning polycyclic aromatic hydrocarbons (PAHs) in Jeddah’s schools, Saudi Arabia, and their implications for health risks to children, is scarce. Classroom air conditioner filter dusts were collected from primary schools in urban, suburban and residential areas of Jeddah. This study aimed to assess the characteristics of classroom-dust-bound PAHs and the health risks to children of PAH exposure. Average PAH concentrations were higher in urban schools than suburban and residential schools. Benzo (b)fluoranthene (BbF), benzo(ghi)perylene (BGP), chrysene (CRY) and Dibenz[a,h]anthracene (DBA) at urban and suburban schools and BbF, BGP, fluoranthene (FLT) and indeno (1, 2, 3, −cd)pyrene (IND) at residential schools were the dominant compounds in classroom dust. PAHs with five aromatic rings were the most abundant at all schools. The relative contribution of the individual PAH compounds to total PAH concentrations in the classroom dusts of schools indicate that the study areas do share a common source, vehicle emissions. Based on diagnostic ratios of PAHs, they are emitted from local pyrogenic sources, and traffic is the significant PAH source, with more significant contributions from gasoline-fueled than from diesel cars. Based on benzo[a]pyrene equivalent (BaP_equi_) calculations, total carcinogenic activity (TCA) for total PAHs represent 21.59% (urban schools), 20.99% (suburban schools), and 18.88% (residential schools) of total PAH concentrations. DBA and BaP were the most dominant compounds contributing to the TCA, suggesting the importance of BaP and DBA as surrogate compounds for PAHs in this schools. Based on incremental lifetime cancer risk (ILC_ingestion_, ILCR_inhalation_, ILCR_dermal_) and total lifetime cancer risk (TLCR)) calculations, the order of cancer risk was: urban schools > suburban schools > residential schools. Both ingestion and dermal contact are major contributors to cancer risk. Among PAHs, DBA, BaP, BbF, benzo(a)anthracene (BaA), benzo(k)fluoranthene (BkF), and IND have the highest ILCR values at all schools. LCR and TLCR values at all schools were lower than 10^−6^, indicating virtual safety. DBA, BaP and BbF were the predominant contributors to cancer effects in all schools.

## 1. Introduction

Indoor and outdoor air pollution, both created by human activities and naturally occurring, is considered a serious problem worldwide, since it has been linked with numerous environmental effects and is responsible for increased mortality and morbidity rates [1,2]. Globally, air pollution is responsible for 6.4 million deaths in 2015, which account for 11% of global deaths [3]. Some persistent air pollutants, over long periods, can accumulate in various environmental media, consequently affecting both humans and animals through inhalation, dermal and ingestion routes of exposure. The most important sources which can contribute to air pollution include energy, transportation, industry, power stations, agriculture activities, households activities and waste management [1]. 

Polycyclic aromatic hydrocarbons (PAHs) are long-lived and widely distributed in the environment due to their physicochemical properties, constituted of two or more aromatic rings and formed mainly during pyrolysis and incomplete combustion processes [4,5]. They are distributed in both gaseous and particulate phases in the atmosphere [6,7]. Low molecular weight (LMW) PAHs are more volatile and exist in the gas phase [8]. High molecular weight (HMW) PAH compounds, which exist mainly as particulates [8], are not easily degraded under normal status and consequently their persistence is increased [9]. There is a little information on technologies for PAH removal from the environment [6,10]. Some PAHs have toxic, carcinogenic, and mutagenic properties [11,12,13,14,15]. According to the International Agency for Research on Cancer [12,16], benzo[a]pyrene (BaP) is classified as carcinogenic to humans (group I), benz[a]anthracene (BaA) and DBA as probable carcinogens (group 2A), and NA, CRY, BbF, BkF and IND as possible carcinogens (group 2B). Previous studies have reported that human exposure to PAH compounds causes toxic effects on cardiorespiratory, reproductive, immune, and developmental processes [11], limits intrauterine growth [13], and is endocrine-disrupting [14].

PAHs originate from both natural and anthropogenic sources [6,17,18]. Burning of fossil fuels, petroleum, wood, coal tar, and gas, which represents the energy production sector, constitutes the main anthropogenic sources of PAHs worldwide, whereas volcanic eruptions and forest fires are the main natural PAH sources [6,19,20,21]. With regard to indoor environments, PAH levels are influenced by human activities and indoor sources, such as tobacco smoke, cooking or heating processes, especially developing ones, which predominantly use open combustion of wood and alternative combustion matter like coal, agricultural remains and dung for cooking and/or heating purposes, decorative candle and incense burning, alongside the penetration of outdoor PAHs, in both particulate and vapor phases, into indoor environments through windows, doors, cracks and ventilation system [6,22,23,24,25,26,27,28,29].

School indoor air quality is considered an important scientific issue concerning children, particularly in primary schools because children spend much of their daily time in schools with higher inhalation rates and higher sensitivity to pollutants. Previous studies reported that indoor air pollutants in schools, even at low levels, cause several health complaints, loss of output, effects on the academic performance [10] and the mental stability of children was affected [30]. PAHs are one of the most significant pollutants in schools due to their health-relevant effects, particularly on children. This is because they produce a larger intake of harmful pollutant compounds compared with adults, due to their faster inhalation and increased physical activity [31,32]. Moreover, the exposure to genotoxic carcinogenic pollutants at of young age people may lead to an elevated risk of cancer in adult life because of genetic disruptions from exposure, like mutations, sister and chromatid exchanges [33,34]. Current published data on the status, sources and health risks of PAHs in schools is scarce and still very limited. The available published data in primary schools come from previous studies in some European cities [4,35], Asia [36,37], and the USA [38].

Indoor air quality is affected by air conditioning (AC), which is widely used in buildings in Saudi Arabia as an effective means to manage indoor temperature and ventilate the building, and otherwise affect air quality in indoor environments. Indoor suspended particulate matter, which can be considered one of the most important indoor pollutants that contain organic and inorganic contaminants, with particle size less than 100 µm, can be attached and deposited on the AC filter during air current impact [39,40]. The settled dust/particles on the AC filter can be resuspended and attached onto indoor roofs, and, thereafter, residents, especially children, can be exposed to them through the ingestion pathway [41]. Therefore, the settled dust on the AC filter can be considered as an exposed dust that has penetrated from the outdoors, through doors, ventilation systems, windows and AC filters for fresh air [42] to the indoor environment, in addition to the suspended and resuspended settled dust from indoor particulate sources and resident activities [43,44]. Therefore, assessing the contamination levels and health risk of pollutants in deposited dust on AC filters is very important, since it can improve our understanding of pollutants in the indoor environment. The main objectives of this study are: (1) to investigate the concentration levels and profiling of PAHs in AC filter dust of some primary schools in Jeddah city that are representative of different environments (urban, residential, suburban); (2) to identify the potential sources of the measured PAH compounds using diagnostic ratios analysis; and (3) to calculate the health risk assessment for the exposed children by using the measured values for PAHs in AC filter samples, toxicity equivalency factors (TEF) and USEPA equations.

## 2. Materials and Methods 

### 2.1. Sampling Sites and Collection

Classroom air conditioner filter dust samples, which represent all of the exposed dust that has migrated to the indoor environment, were collected from primary schools in urban, suburban and residential areas of Jeddah (Figure 1), the second largest city of Saudi Arabia. Almost all the air pollutants near the sampling sites arise from the emissions from traffic activities, since more than 1.4 million vehicles, fueled mainly by unleaded gasoline and diesel, run in Jeddah’s streets. Full details of the study area and its climatic characteristics, sampling sites and the methods of sampling collection and preparation have been reported elsewhere [45]. In Brief, 40 classrooms air conditioner filter dust samples were collected from 10 primary schools situated in different areas during the spring season (April–May 2019). This sampling period was predefined to obtain maximum loadings of particulate matter during two months of the spring season. At each school, four classrooms air conditioner filter dust samples were collected simultaneously from different classrooms from the air conditioner filters of each classroom. A plastic brush, clean polythene sheets and airtight polyethylene bags were used for sample collection. The collected samples were stored in clean-labelled polyethylene bags, air-dried at room temperature in the laboratory, and then the coarse impurities were removed using a 1.0 mm mesh nylon sieve. The remainder of the dust samples were homogenized and sieved through a 38-μm sieve size and kept in small self-sealing plastic bags for analysis.

### 2.2. Extraction of PAHs in Classroom Dust

Classroom dust-bound PAH compounds were extracted from AC dust samples using a Soxhlet apparatus. They were extracted with a mixture of dichloromethane/n-hexane/acetone (1/1/1, v/v/l), for 16h according to [46]. A rotary evaporator was used to concentrate the organic extract, and cleaned with clean columns containing anhydrous sodium sulfate, silica gel, alumina, sand, and glass wool [47,48,49]. The extracted eluent from the clean-up procedure were concentrated using a rotary evaporator and exchanged to 1 mL hexane, and then stored frozen until analysis. 

### 2.3. Analysis of PAHs in Classroom Dust

For classroom dust-bound PAH compound analysis, one microliter (μL) of extract was withdrawn from the samples, including the blank samples, and injected into a Hewlett–Packard (HP6890, Agilent, Santa Clara, CA, USA) gas chromatograph (GC), fitted with a flame ionization detector (FID). An HP-5 capillary column was used with hydrogen as the carrier gas. An external standard solution was used to quantify the concentration of PAH compounds in classroom AC filter dust. Full details of the quality assurance/quality control (QA/QC) process used to check the accuracy of the obtained results have been reported elsewhere [50]. Briefly, quality assurance/quality control (QA/QC), including GC/FID calibration and detection limits, reagent blanks, analytical standards and standard spike recoveries, was employed to check the reliability of the obtained PAH results. The GC/FID was checked daily using retention times and responses of PAH compounds in the standard calibration mixture injection. The concentrations of the target PAH compounds were quantified using an external standard calibration. To prepare the calibration solution, PAH mixture standards containing 16 compounds (Supelco, Inc., St. Louis, MO, USA) were used. These standard solutions were used to produce calibration curves and were analyzed with the samples. The relative standard deviation of the replicated analyses of the calibration standard ranged from 4% to 6% for the 16 measured PAH compounds. The average recovery of each PAH ranged from 70% to 108%. The limit of detection for the measured individual PAH compounds varied from 0.05 ng/g to 0.38 ng/g. PAHs including naphthalene (NA), acenaphthylene (ACY), acenaphthene (ACE), fluorene (FLU), phenanthrene (PHE), anthracene (ANT), fluoranthene (FLT), pyrene (PYR), benzo(a)anthracene (BaA), chrysene (CRY), benzo(b)fluoranthene (BbF), benzo(k)fluoranthene (BkF), benzo(a)pyrene (BaP), dibenzo(a,h)-anthracene (DBA), benzo(ghi)perylene (BGP), and indeno (1, 2, 3, −cd)pyrene (IND) were the target PAH compounds in the classroom AC filter dust. 

PAH compounds in the classroom AC filter dust were classified into two rings (NA), three rings (ACY, ACE, FLU, PHE and ANT), four rings (FLT, PYR, BaA and CRY), five rings (BbF, BaP and DBA) and six rings (IND and BGB) depending on the number of aromatic rings. They were also classified as low molecular weight (LMW) PAH compounds (2–3 aromatic rings) and high molecular weight (HMW) PAH compounds (4–6 aromatic rings).

### 2.4. Sources Identification of PAHs in Classroom Dust

PAHs can be emitted in the atmosphere from different emission sources, and, consequently, their chemical composition patterns are significantly varied. To understand the fate and transport of PAHs in classroom environments, it is very important to identify the possible sources of PAHs. Therefore, some PAH isomeric concentration ratios can be used to identify different possible sources of PAHs in the environment. For example, some concentration ratios of the measured PAH compounds in classroom AC filter dust, like ANT/(ANT+PHE), FLU/(FLU + PYR), FLT/PYR, IND/(IND + BGP), PHE/ANT, BaA/CRY, BaP/BGP and LMW-PAHs/HMW-PAHs were used to identify the possible sources of emission of PAHs [27,51,52,53,54,55,56,57,58,59,60,61,62,63,64,65,66].

### 2.5. Health Risk Assessment of PAHs in Classroom Dust

Toxicity equivalency factors (TEF), carcinogenic PAH (CPAHs) determination, and incremental lifetime cancer risk (ILCR) were used in the present study to evaluate the potential risks of PAH compounds in the classroom dust of the primary schools [67,68,69,70,71,72,73].

#### 2.5.1. Carcinogenic Potency of PAHs (BaPequi) in Classroom Dust

The carcinogenic potency of the measured PAH compounds in the classroom AC filter dust of the study primary schools was calculated as benzo(a)pyrene equivalence (BaP_equi_) according to the TEFs proposed by [70]. Among the PAH compounds, BaP was the most toxic compound, selected as a reference chemical, and was assigned a value of one [74,75]. The following Equation (1) was used in calculation of BaP_equi_:BaP_equi_ (ng/g) = C × TEF(1)
where C is the concentration of individual PAH compounds in classroom AC filter dust; TEF is the corresponding individual equivalency factor for PAH compounds proposed by Nisbet and La Goy [70]. Thereafter, the cancer potency of the total PAH compounds in classroom AC filter dust was calculated by the summation of estimated cancer potency relative to BaP for all PAH compounds.

#### 2.5.2. Carcinogenic Estimation of PAHs in Classroom Dust

Carcinogenic estimation for the probable carcinogenic PAH compounds was calculated as the percentage of individual carcinogenic PAH compounds to the total PAHs [68]. From the measured PAHs in classroom AC filter dust, carcinogenic PAHs including BaA + BbF + BkF + BaP + DBA + IND were used in the calculation according to the International Agency for Research on Cancer [76] and USEPA [77].

#### 2.5.3. Incremental Lifetime Cancer Risk (ILCR) from Exposure to PAHs in Classroom Dust

In the present study, health risk assessment for schoolchildren exposed to classroom dust-bound PAHs was evaluated using risk assessment standard models developed by the Environmental Protection Agency (EPA) of the United States [67,69,71,73,78]. Direct inhalation, ingestion, and dermal contact are the main three exposure pathways for children in the primary schools, which exposed them to PAHs in classroom dust. The incremental lifetime cancer risk (ILCR) is one way of estimating the long-term exposure risk for school children associated with exposure to PAHs through these different exposure pathways. For each PAH compound in the classroom dust, carcinogenic risk was calculated by the summation of the individual risks calculated for the different three exposure pathways. Then, the total PAH carcinogenic risk was calculated by summing the individual PAH risks for each PAH compounds for the three exposure pathways. BaP equivalence (BaP_equi_) concentration was used in the calculation of risk assessment. The following Equations (2)–(5) were used in the calculation of ILCR in terms of direct inhalation, ingestion and dermal contact:(2)$${\;\mathrm{ILCR}}_{\mathrm{ingestion}}=\frac{\mathrm{Cs}\;\times \left\{{\mathrm{CSF}}_{\mathrm{ingestion}}\times \sqrt[{3}]{\left(\frac{\mathrm{BW}}{70}\right)}\right\}\times {\;\mathrm{IR}}_{\mathrm{ing}.}\times \;\mathrm{EF}\;\times \;\mathrm{ED}}{\mathrm{BW}\;\times \;\mathrm{AT}\;\times 10^{6}\;}$$
(3)$${\mathrm{ILCR}}_{\mathrm{inhalation}}=\frac{\mathrm{Cs}\;\times \left\{{\mathrm{CSF}}_{\mathrm{inhalation}}\times \sqrt[{3}]{\left(\frac{\mathrm{BW}}{70}\right)}\right\}\times {\;\mathrm{IR}}_{\mathrm{inhal}.}\times \;\mathrm{EF}\;\times \;\mathrm{ED}}{\mathrm{BW}\;\times \;\mathrm{AT}\;\times \;\mathrm{PEF}\;}$$
(4)$${\mathrm{ILCR}}_{\mathrm{dermal}}=\frac{\mathrm{Cs}\;\times \left\{{\mathrm{CSF}}_{\mathrm{dermal}}\times \sqrt[{3}]{\left(\frac{\mathrm{BW}}{70}\right)}\right\}\times \;\mathrm{SA}\;\times \;\mathrm{AF}\;\times \;\mathrm{ABS}\;\times \;\mathrm{EF}\;\times \;\mathrm{ED}}{\mathrm{BW}\;\times \;\mathrm{AT}\;\times 10^{6}}$$
(5)$$\mathrm{Carcinogenic}\;\mathrm{risk}={\mathrm{ILCR}}_{\mathrm{ingestion}}+{\mathrm{ILCR}}_{\mathrm{inhalation}}+{\mathrm{ILCR}}_{\mathrm{dermal}}$$
where the ILCR is incremental lifetime cancer risk; CS represents the concentration of the PAH compounds (mg BaP_equi_/kg) in classroom AC filter dust based on toxic equivalent of BaP using the TEF; CSF_ingesion_, CSF_inhelation_, CSF_dermal_ are the carcinogenic slope factors (mg/kg/day) for ingestion, inhalation and dermal contact pathways, respectively; BW is body weight (kg); AT is average life span (years); EF represents the exposure frequency (day/year); ED is the exposure duration (years); IR_inhalation_ represents the inhalation rate (m^3^/day); IR_ingestion_ is the soil intake rate (mg/day); SA represents the dermal surface exposure (cm^2^); AF is the dermal adherence factor (mg/cm^2^/h); ABS represents the dermal adsorption fraction; and PEF is the particle emission factor (m^3^/kg). In the present study, full details information about the values of exposure factors for children [71,78,79,80,81,82,83,84] applied to the above models (Equations (2)–(5)) are given in Table 1.

## 3. Results and Discussion

### 3.1. Classroom Dust-Bound PAH Concentrations

The mean concentrations of the individual PAH compounds in the classroom AC filter dusts collected from the different primary schools of Jeddah are shown in Figure 2. The individual PAH concentrations, in descending order, were: BGP ≈ BbF > CRY > DBA > BaP > FLT > BaA > IND > BkF > PYR > PHE > ANT > NA > FLU > ACE >ACY. The concentration values were 54.24 ± 41.40 ng/g for NA, 32.10 ± 20.72 ng/g for ACY, 37.83 ± 20.56 ng/g for ACE, 42.71 ± 20.97 ng/g for FLU, 68.57 ± 30.67 ng/g for PHE, 55.25 ± 21.84 ng/g for ANT, 162.34 ± 53.33 ng/g for FLT, 87.90 ± 39.77 ng/g for PYR, 154.59 ± 70.38 ng/g for BaA, 187.89 ± 82.02 ng/g for CRY, 248.24 ± 70.02 ng/g for BbF, 132.87 ± 32.62 ng/g for BkF, 163.87 ± 68.53 ng/g for BaP, 179.40 ± 84.12 ng/g for DBA, 148.51 ± 54.45.88 ng/g for IND, and 248.39 ± 37.65.39 ng/g for BGP. Based on aromatic ring number, the concentration of the PAH compounds with five aromatic rings represented the highest levels, followed by four aromatic rings, six aromatic rings, three aromatic rings and two aromatic rings in classroom AC filter dust from Jeddah schools (Figure 2). The average concentration values were 54.24 ± 41.40 ng/g, 236.41 ± 111.16 ng/g, 592.72 ± 236.71 ng/g, 724.10 ± 251.81 ng/g and 396.91 ± 88.28 ng/g for two, three, four, five and six aromatic rings, respectively. Among all schools, there are no specific indoor emission sources or activities that could justify the levels of PAHs. Therefore, the sources of PAH compounds in classroom AC filter dust mostly originated from the outdoor environment, since the traffic that circulates around the Jeddah schools is the outdoor anthropogenic source of dust-bound PAHs [85,86]. Indoor levels of particles might result from infiltration of outdoor particles to indoor air [87,88,89,90]. Therefore, the PAH levels in classroom AC filter dust from Jeddah schools may be lower, higher or similar to those found in other city schools of the world. This may be attributed to variation in the main sources of these PAHs in the outdoor environment, like traffic density, human activities, and industrial activities. For example, the concentration of total classroom dust-bound PAH compounds in Jeddah primary schools (2004.37 ng/g) was higher than that reported (618.4–1667 ng/g) in random schools located close to downtown and to the oil and gas industrial area of Kuwait [91] and lower than in schools of the Al-Ahmadi province, Kuwait (3650 ng/g) [92].

### 3.2. Spatial Variations in Classroom Dust-Bound PAH Concentrations

Spatial differences in the average concentrations of the individual particle-bound PAH compounds in collected classroom AC filter dust from urban, suburban and residential schools in Jeddah are shown in Figure 3. The average concentrations of the individual PAH compounds were higher in urban schools than in suburban and residential schools. The most abundant classroom dust-bound PAHs were BbF, BGP, CRY and DBA at urban and suburban schools and BbF, BGP, FLT and IND at residential schools. The average concentrations of individual PAH compounds ranged from 55.34 ng/g (ACY) to 324.89 ng/g (BbF) at urban schools, from 23.27 ng/g (ACE) to 250.0 ng/g (BGP) at suburban schools and from 15.35 ng/g (ACY) to 193.16 ng/g (BGP) at residential schools. Based on aromatic ring number, the classroom dust-bound PAHs with five aromatic rings were the most abundant, followed by four aromatic rings, six aromatic rings, three aromatic rings and two aromatic rings at the different schools (Figure 4). The average concentrations were 985.5, 703.7 and 483.1 ng/g for five aromatic rings, 843.9, 560.6 and 373.7 ng/g for four aromatic rings, 493.9, 373.4 and 321.4 ng/g for six aromatic rings, 357.1, 214.1 and 138.1 ng/g for three aromatic rings and 101.3, 37.7 and 12.6 ng/g for two aromatic rings at urban, suburban and residential schools, respectively. Based on the concentrations of ∑PAH compounds in each schools of different areas, they could be classified as follows: urban schools > suburban schools > residential schools. The concentrations of the ΣPAH compounds were 2781.7, 1891.4 and 1339.9 ng/g at urban, suburban and residential schools, respectively. Vehicular traffic emissions are the main sources of particle-bound PAH compounds in the outdoor environment [93,94,95,96]. The wide differences in the level of PAH compounds among schools in different areas result from the differences in traffic emissions due to the differences in traffic density, since the relatively higher traffic density around the urban schools compared with suburban and residential schools leads to an increase in the emission of PAHs in outdoor and then consequently in the indoor environment of urban schools through penetration.

### 3.3. Distribution of Classroom Dust-Bound PAHs

The relative contributions of the individual dust-bound PAH compounds to the total dust-bound PAHs concentrations in classroom AC filter dust from urban, suburban and residential schools were nearly similar (Figure 5). This similarity in contributions of the individual PAH compounds indicates that the study areas do share a common source of vehicle emission [78,97]. Generally, the contribution of the individual dust-bound PAH compounds increased with increasing molecular weight of the PAH compounds at the different schools, with highest values for BbF and BGP at all schools and lowest values for ACY at urban and residential schools and ACE at suburban schools. Based on aromatic ring number, the distribution of the dust-bound PAHs with four to six aromatic rings (83.52% at urban schools, 86.70% at suburban schools and 87.93% at residential schools) was predominant (Figure 5). This result is in agreement with previous studies that reported that the distribution of dust-bound PAH compounds depends on molecular weight [98,99], and compounds with four to six PAH aromatic ring numbers were predominant [100,101]. The predominance of high-molecular-weight PAH compounds in classroom AC filter dust in the present study at the different schools indicates that a significant fraction of dust-bound PAHs mostly come from pyrogenic sources [102], such as petroleum fuel combustion [103] and vehicular emissions [104]. The PAH speciation in gasoline vehicle soot is enriched by heavy molecular weight PAHs [105].

### 3.4. Possible Sources of Classroom Dust-Bound PAHs

Diagnostic ratios of individual PAH congeners, like ANT/(ANT+PHE), FLU/(FLU + PYR), FLT/PYR, IND/(IND + BGP), PHE/ANT, BaA/CRY, BaP/BGP and LMW-PAHs/HMW-PAHs, can provide insight regarding the possible origin of the PAHs. The ratios that were calculated for classroom dust-bound PAHs at urban, suburban and residential schools are presented in Table 2. The ratios of the LMW-PAHs/HMW-PAHs indicate whether the origin of PAHs is pyrogenic (ratio < 1) or petrogenic (ratio > 1) [4,60]. The values of these diagnostic ratios at all schools were lower than one, indicating a pyrogenic source of PAHs. Similarly, the ratios of PHE/ANT (< 10) and FLT/PYR (> 1) indicate a pyrogenic source of PAHs at all schools. The ANT/(ANT + PHE) ratios at all schools were higher than 0.1, which suggests a combustion source [64]. Traffic emissions were the most probable source of PAHs at these schools. Diagnostic BaA/CRY ratio were higher than 0.35 at all schools and thus indicated and confirmed vehicular emission and/or fuel combustion [4]; the diagnostic IND/IND +BGP values at all schools also suggested vehicular (both diesel and gasoline) emissions [27,53,54,55,56,57,58,61]. To differentiate between the exhaust origin, FLU/FLU +PYR can be used with values higher than 0.5 for diesel emissions and lower than 0.5 for gasoline type [27]. The values of the FLU/FLU +PYR ratio in the present study were lower than 0.5, suggesting an effect of gasoline emissions at all schools [59]. The IND/(IND +BGP) values in this study ranged from 0.33–0.40 at all schools, suggesting the effect of road dust around the schools [27,53,54,55,56,57,58]. At all schools, diagnostic ratios of BaA/BaA + CRY ranged from 0.43–0.47, indicating emissions mostly from gasoline-fueled cars [59]. FLT, PYR, BaA, CRY, BbF, BaP, IND and BGP are combustion-related non-alkylated PAH compounds (CPAHs). The diagnostic ratio of the total concentrations of CPAHs to the total concentration of PAHs (∑CPAHs/∑PAHs) is used to distinguish possible emissions of PAHs from mobile combustion versus stationary sources [61]. A CPAHs/∑PAHs concentration ratio less than one indicates that the PAH compounds come from mobile sources. In the present study, the CPAHs/∑PAHs concentration ratios were 0.74 at urban schools, 0.78 at suburban schools and 0.80 at residential schools. This result suggests that mobile emissions are the principal source of classroom dust-bound PAH compounds in Jeddah schools.

It is necessary to point out that the diagnostic ratio of more-reactive PAH to less-reactive PAH was used to give details about the PAH source, degradation of PAHs under the effect of photochemical reactions and the aging of the air mass [106,107]. For example, a BaA/CRY concentration ratio higher than 0.40 suggests that the pollution is freshly emitted and photochemical processing of the air mass is relatively low, whereas a ratio of lower than 0.40 indicates that the main PAH sources are not local or the air masses are aged [106]. BaA/CRY concentration ratios in the present study were 0.82 at urban schools, 0.87 at suburban schools and 0.75 at residential schools (Table 2), which indicate that PAHs in classroom dust are freshly emitted from local sources around the schools.

Based on the applied diagnostic ratios analysis, it is possible to assume that vehicular traffic is a significant emission source for PAHs present in classroom dust, with significant contributions from gasoline-fueled rather than diesel cars. Moreover, classroom dust-bound PAH compounds are emitted from a pyrogenic source and they are freshly emitted from a local source around the schools.

### 3.5. Health Risk Assessment of Classroom Dust-Bound PAH Compounds 

#### 3.5.1. Carcinogenic Potency of PAHs (BaP_equi_)

Regarding to BaP, it is considered one of the most important PAH compounds due to its potent carcinogenic properties and is considered a sufficient indicator of the carcinogenicity of the total PAH compounds [108,109,110]. The BaP compound was also used widely as an air quality indicator, and a marker for total PAH exposure and overall carcinogenicity for these PAHs [111,112,113]. Harrison et al. [114] reported that around 88% of BaP was evaluated to come from ambient traffic in British cities. Therefore, it can usually be used as a key indicator for assessment of the health risk of PAHs in traffic areas.

Table 3 shows the carcinogenic potencies of the individual PAH compounds in the classroom AC filter dust at the study primary schools, which were calculated as benzo(a)pyrene equivalence (BaP_equi_) according to the TEFs proposed by Nisbet and LaGoy [70]. The concentrations of BaP_equi_ for PAH compounds were calculated by multiplying the concentration of individual PAH compounds by their corresponding TEF values [70,115]. Then, the cancer potency for each individual PAH compound was evaluated based on its corresponding calculated BaP_equi_ concentration value. In the present study, the concentrations of BaPeq for the individual classroom bound PAH compounds ranged from 0.06 ng BaPequi/g for ACY, ACE and FLU to 267.54 ng BaP_equi_/g for DBA at urban schools, 0.02 ng BaP_equi_/g for ACE to 170.69 ng BaP_equi_/g for DBA at suburban schools, 0.02 ng BaP_equi_/g for NA, ACY and FLU to 99.97 ng BaP_equi_/g for DBA at residential schools and 0.03 ng BaP_equi_/g for ACY to 179.40 ng BaP_equi_/g for DBA at all schools (Table 3). Moreover, the total carcinogenic activities (TCA) of the total PAHs in classroom dust were 600.6, 397.0, 252.9 and 416.8 ng BaP_equi_/g at urban, suburban, residential and all schools, respectively. They represented 21.59% (urban schools), 20.99% (suburban schools), 18.88% (residential schools) and 20.79% (all schools) of the total PAHs concentrations. Regarding the relative contribution of the individual carcinogenic activity for BaA, BbF, BkF, BaP, DBA, CRY and IND to the TCA (Table 3), DBA and BaP were the most dominant compounds; they accounted for 44.54% and 39.31% (urban schools), 42.99% and 38.98% (suburban schools), 39.52% and 39.50 (residential schools) and 43.04% and 39.25% (all schools) from the TCA, respectively. In spite of the fact that BaP and DBA, as surrogate compounds for PAHs in school classroom environments, are significantly important, other compounds, like BbF, BaA, IND and BkF, also play a role in the TCA.

#### 3.5.2. Carcinogenic PAH Estimation

The total concentrations of carcinogenic PAH compounds (BaA + BbF + BkF + BaP + DBA + IND) in the classroom dust-bound PAHs were 1400.2 ng/g (urban schools), 987.89 ng/g (suburban schools), 693.51 ng/g (residential schools), and 1027.20 ng/g (all schools). These concentrations accounted for 50.3%, 52.3%, 51.8% and 51.2% of the total PAH concentrations at urban, suburban, residential and all schools, respectively. The relative contributions of the individual carcinogenic PAH compounds to the total concentrations of carcinogenic PAHs were 15.91%, 16.10%, 11.85% and 15.05% for BaA, 23.20%, 23.52%, 27.03%, and 24.17% for BbF, 11.21%, 14.77%, 13.81%, and 12.93% for BkF, 16.86%, 15.67%, 14.40%, 15.93%, and 15.93% for BaP, 13.71%, 12.69%, 18.49%, and 14.46% for IND, and 19.11%, 17.28%, 14.42%, and 17.47% for DBA at urban, suburban, residential and all schools, respectively. Generally, BbF was the dominant carcinogenic compound at all schools.

#### 3.5.3. Incremental Lifetime Cancer Risk (ILCR)

To identify the potential cancer risks to children exposed to classroom dust-bound PAHs, the ILCR for the ingestion (ILC_ingesion_), inhalation (ILCR_inhalation_), and dermal (ILCR_dermal_) pathways needs to be calculated, based on the values of toxic equivalent of BaP (TEF) and cancer slope factor (CSF). ILCR values of ≤10^−6^ indicate virtual safety, values between 10^−6^ and 10^−4^ suggest potential risk, and values more than 10^−4^ suggest a potentially high risk [69,116]. The ILCR assessment of the individual PAH compound concentrations in the classroom dust of different schools through ingestion, inhalation, and dermal contact pathways was performed for children (Figure 6, Figure 7 and Figure 8). Based on the calculated ILCR values from individual and ∑PAHs exposure in classroom dust, the sequence of cancer risk (ILC_ingestion_, ILCR_inhalation_, ILCR_dermal_ and total lifetime cancer risk (TLCR)) was urban schools > suburban schools > residential schools. This variation in cancer risks between the schools indicates large effects of outdoor source emissions of PAHs on the health risk to children. Among the PAH compounds, DBA, BaP, BbF, BaA, BkF, and IND showed high ILCR_ingestion_, ILCR_inhalation_ and ILCR_dermal_ values compared with other PAH compounds in the different schools and in all schools. The values of LCR and TLCR were in the order of ingestion > dermal contact > inhalation pathways at all the different schools. The higher cancer risk values for direct ingestion than the corresponding risk values for dermal contact and inhalation may be attributed to hand-to-mouth activity in children. This is in agreement with previous studies [67,69,78] which reported that young children were the most sensitive subpopulation due to their hand-to-mouth activity. In the present study, the values of LCR and TLCR were lower than 10^−6^, indicating virtual safety at all schools.

The mean relative contributions of individual PAH compound cancer risk to total PAHs cancer risks in different schools are represented graphically in Figure 9. It can be seen that, DBA, BaP and BbF were the predominant contributors to cancer effects in the classrooms. DBA, BaP and BbF accounted for 44.62%, 39.38% and 5.42% of the total PAH cancer effects in urban schools, 43.05%, 39.03% and 5.86% in suburban schools, 39.68%, 39.65% and 22.7% in residential schools, and 43.12%, 39.32% and 5.97% in all Jeddah schools, respectively.

## 4. Conclusions

This study provides information dealing with classroom dust-bound PAH levels, characterizing their health risks in the indoor air of primary schools in Jeddah, as there is little data in literature on this issue. A total of 16 PAH compounds were analyzed in classroom air conditioner filter dusts collected from schools located in three different areas, namely urban, suburban and residential areas, in Jeddah. Urban schools that are located in areas with high traffic density exhibited the highest levels of PAHs, whereas the lowest concentrations were found in schools situated in residential areas. The results indicate that the vehicular traffic around the schools in all of the locations was the main source of the PAHs in the indoor air of school classrooms through infiltration of ambient emissions indoors. The individual PAH concentrations in descending order were BGP ≈ BbF > CRY > DBA > BaP > FLT > BaA > IND > BkF > PYR > PHE > ANT > NA > FLU > ACE > ACY in all schools. The relative contributions of the individual PAH compounds to total PAH concentrations in classroom AC filter dust from urban, suburban and residential schools were nearly similar, suggesting that the study areas do share a common source of vehicle emissions. High-molecular-weight PAH compounds were dominant in the classroom dusts of all schools. Diagnostic ratios revealed that pyrogenic sources and local emissions from vehicular fuel combustion, especially gasoline, were the main sources of dust- bound PAHs in classroom environments.

The highest average concentrations of total carcinogenic activity (TCA) for all measured PAHs were found in urban schools, whereas the lowest were found in residential schools. These represent 21.59% (urban schools), 20.99% (suburban schools), 18.88% (residential schools) and 20.79% (all schools) of the total PAH concentrations. DBA and BaP compounds were the most dominant contributors to the TCA, indicating the significance of BaP and DBA as surrogate compounds for PAHs in indoor air of schools located in traffic areas.

Based on the estimated values of ILCR_ingestion_, ILCR_inhalation_, ILCR_dermal_ and cancer risks for children exposed to PAHs in classroom AC filter dusts, the sequence of cancer risk of the different schools was urban schools > suburban schools > residential schools. Dust ingestion and dermal contact were the major exposure pathways for PAH exposure that contributed to cancer risks. DBA, BaP and BbF were the predominant contributors to cancer effects for children in the classrooms of all schools. The values of LCRs and TLCR at all schools were lower than 10^−6^, indicating virtual safety.

It can be concluded that, even in the absence of significant potential health risks to children from exposure to classroom dust-bound PAHs in Jeddah primary schools, the health risks might increase if the dust-bound PAHs levels also increase. Therefore, more studies, including a large number of schools in various areas, are required and should be done regularly to understand the real scale of PAH pollution and its health risks in the schools of Saudi Arabia.

## Figures and Tables

**Figure 1 ijerph-17-02779-f001:**
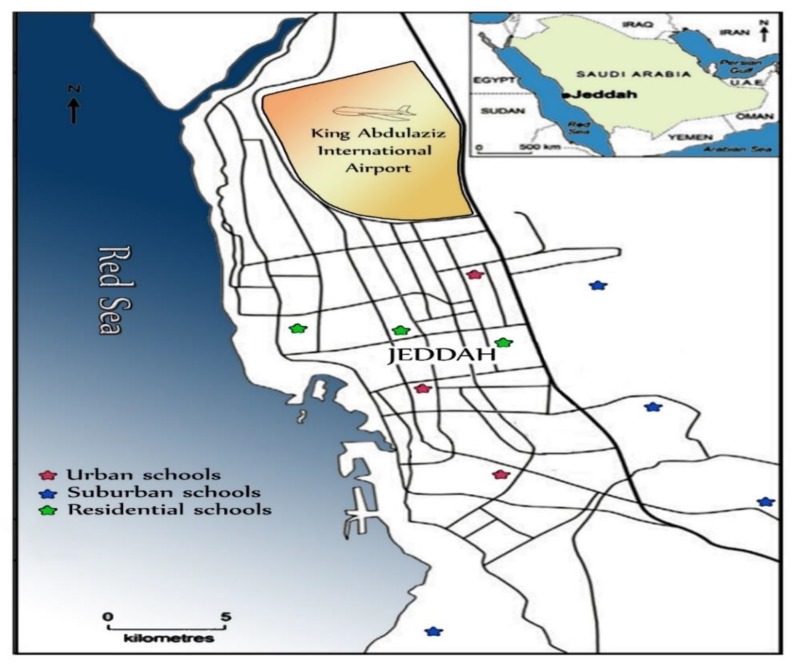
Map of Jeddah showing distribution of primary school sampling sites in the different areas.

**Figure 2 ijerph-17-02779-f002:**
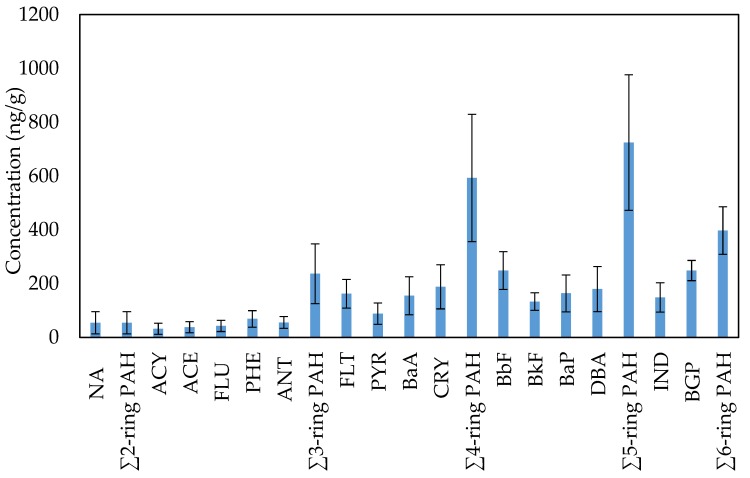
Concentrations of the individual PAH compounds in classroom AC filter dust of all Jeddah schools.

**Figure 3 ijerph-17-02779-f003:**
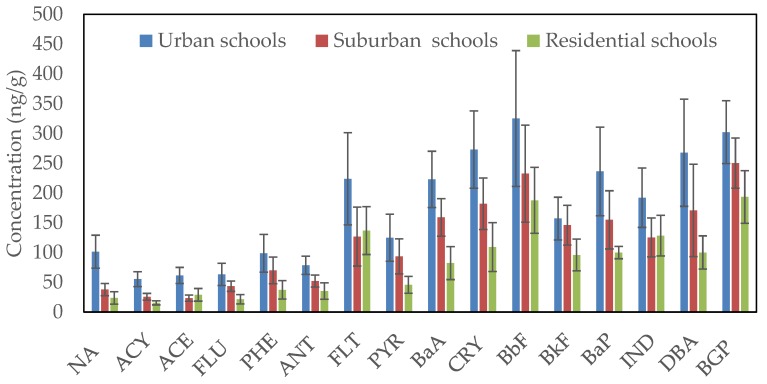
The concentrations of the individual PAH compounds in the classroom AC filter dust of different schools.

**Figure 4 ijerph-17-02779-f004:**
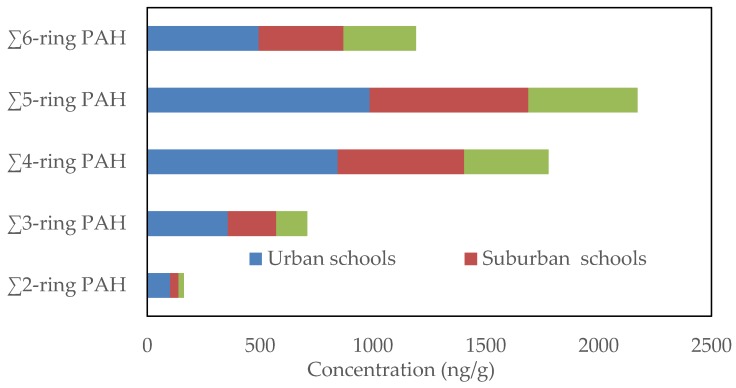
The concentrations of different categories of PAHs based on aromatic ring number in classroom AC filter dust of different schools.

**Figure 5 ijerph-17-02779-f005:**
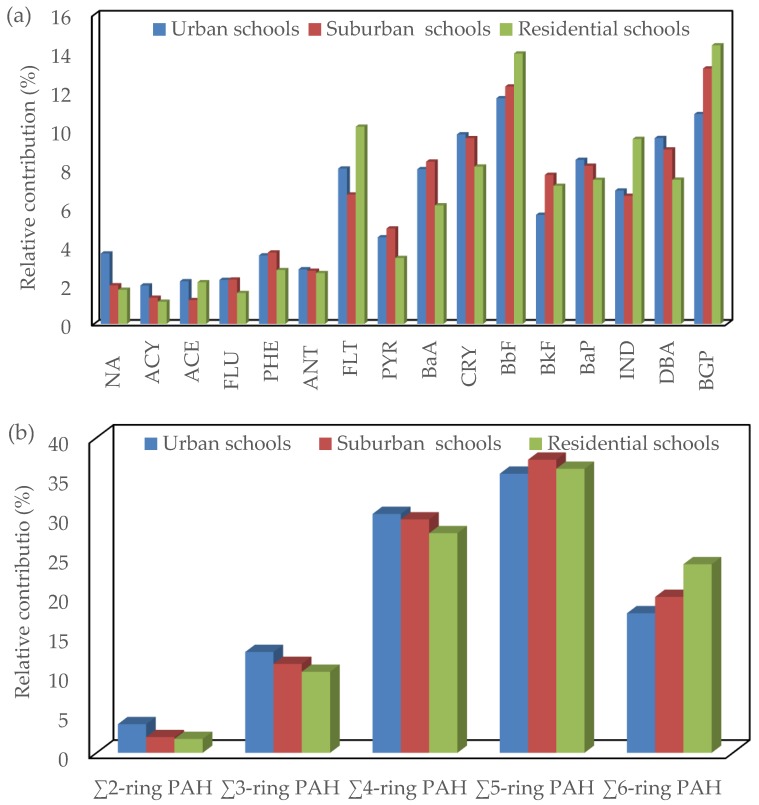
Relative contribution of each individual PAH compound and different categories of PAHs based on aromatic ring number to the total PAHs concentrations in classroom AC filter dust of different schools: (**a**) individual PAH compounds and (**b**) two to six-ring PAHs.

**Figure 6 ijerph-17-02779-f006:**
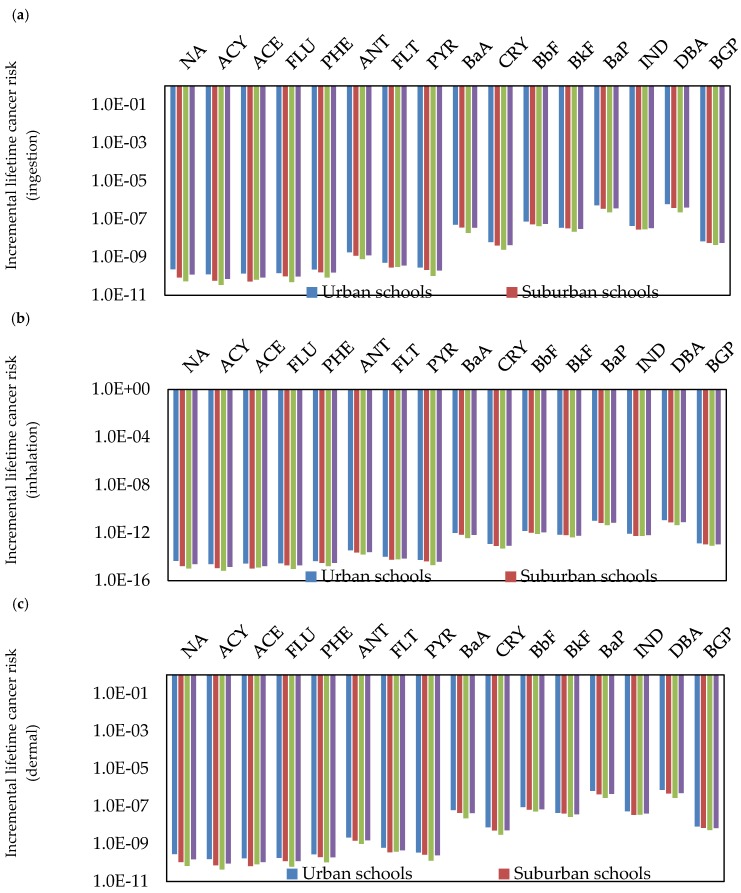
Incremental lifetime cancer risk (LCR) of the individual PAH compound concentrations for children in different schools of Jeddah.

**Figure 7 ijerph-17-02779-f007:**
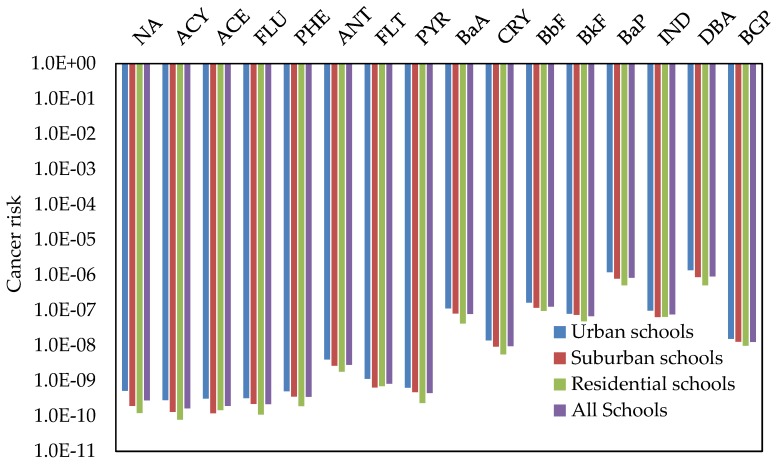
Cancer risk of the individual PAH compound concentrations for children in different schools of Jeddah.

**Figure 8 ijerph-17-02779-f008:**
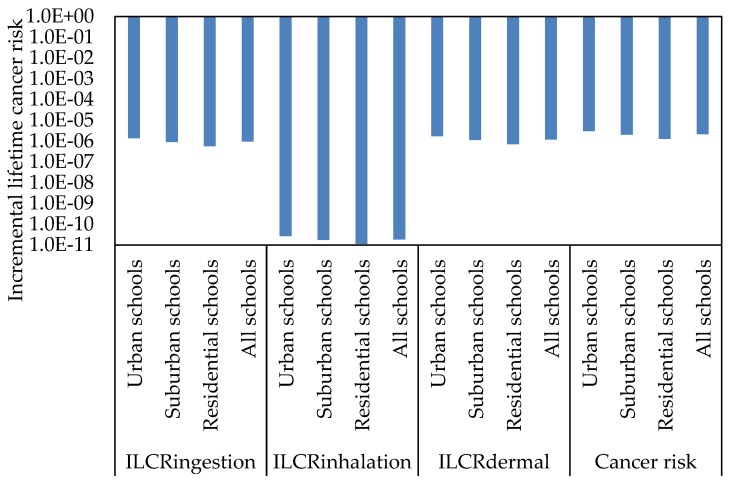
Incremental lifetime cancer risk and cancer risk of the∑PAHs concentrations for children of different primary schools.

**Figure 9 ijerph-17-02779-f009:**
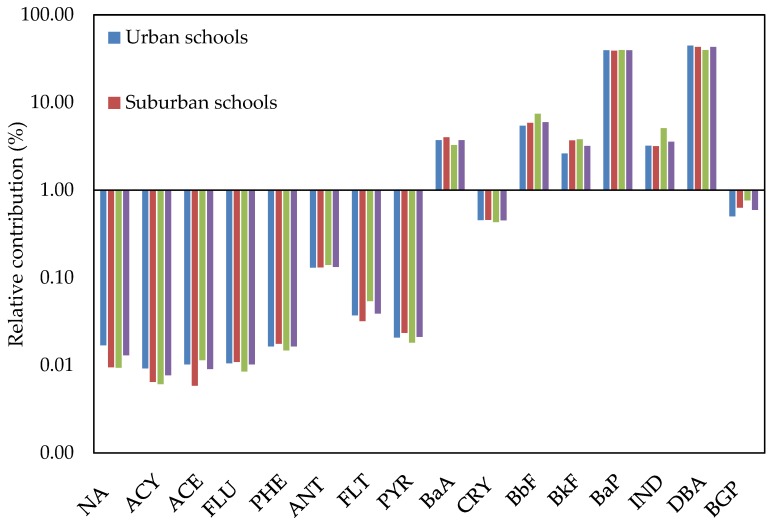
Relative contributions of individual PAH compound cancer risks to total cancer risk in children’s schools.

**Table 1 ijerph-17-02779-t001:** Values of exposure parameters and factors used for incremental lifetime cancer risk assessment.

Exposure Factors	Symbol	Unit	Child	Reference
BaP_equi_. concentration for PAH compounds	Cs	mg BaP_equi_/kg		Present study
Ingestion rate	IR_ingestion_	mg/day	200	[79]
Exposed skin area	SA	cm^2^/day	2800	[79]
Skin adherence factor	SAF	mg/cm^2^	0.2	[79,80]
Exposure frequency	EF	days/year	167	[81]
Exposure duration	ED	year	6	[79]
Body weight	BW	kg	15	[71]
Averaging time (70 years × 365 days/year)	AT	days	25,550	[78,82]
Dermal adsorption fraction	ABS	unitless	0.13	[79]
Inhalation rate	IR_inhalation_	m^3^/day	10	[83]
Particle emission factor	PEF	m^3^/kg	1.36 × 10^9^	[79]
Carcinogenic slope factor for ingestion	CSF_ingestion_	mg/kg/day	7.3	[84]
Carcinogenic slope factor for inhalation	CSF_inhalation_	mg/kg/day	3.85	[84]
Carcinogenic slope factor for dermal	CSF_dermal_	mg/kg/day	25	[84]

**Table 2 ijerph-17-02779-t002:** Diagnostic ratios of classroom dust-bound PAHs at the different schools.

Ratio	Schools			Value	Source	Reference
	Urban	Suburban	Residential			
BaA/CRY	0.82	0.87	0.75	0.28–1.20	Gasoline engines	[51,52]
				0.17–0.36	Diesel engine	
IND/(IND + BGP)	0.39	0.33	0.4	0.21–0.22	Gasoline cars	[27,53,54,55,56,57,58]
				0.35–0.70	Diesel emissions	
				0.56	Coal combustion	
				0.62	Wood combustion	
				0.36	Road dust	
BaP/(BaP + CRY)	0.46	0.46	0.48	0.5	Diesel	[27]
				0.73	Gasoline	
FLU/(FLU + PYR)	0.34	0.32	0.32	>0.5	Diesel	[27]
				<0.5	Gasoline	
BaA/(BaA + CRY)	0.45	0.47	0.43	1.6	Diesel cars	[59]
				0.33	Gasoline cars	
				2.18	Wood combustion	
∑PAHs_LMW_/∑PAHs_HMW_	0.2	0.15	0.14	>1	Petrogenic	[60]
				<1	Pyrogenic	
∑CPAHs/∑PAHs	0.74	0.78	0.8	<1	Mobile source	[61]
BGP/BaP	1.28	1.61	1.93	1.2–2.2	Diesel cars	[61,62,63]
				2.5–3.3	Gasoline cars	
				0.86, 0.91	Road dust	
ANT/(ANT + PHE)	0.44	0.43	0.49	<0.1	Petroleum source	[64]
				>0.1	combustion source	
PHE/ANT	1.26	1.34	1.06	<10	Pyrogenic	[65]
				>15	Petrogenic	
FLT/PYR	1.79	1.36	2.99	>1	Pyrogenic	[65]
				<1	Petrogenic	
IND/BGP	0.64	0.5	0.66	0.4	Gasoline engine	[66]
				≈1	Diesel engine	

**Table 3 ijerph-17-02779-t003:** Concentrations and BaP equivalent concentrations for PAH compounds in classroom AC filter dust at all schools.

PAHs	TEF	Concentration (Urban Schools)		Concentration (Suburban Schools)		Concentration (Residential Schools)		Concentration (All Schools)	
		ng/g	ng BaP_aquiv_/g	ng/g	ng BaP_aquiv_/g	ng/g	ng BaP_aquiv_/g	ng/g	ng BaP_aquiv_/g
NA	0.001	101.34	0.10	37.73	0.04	23.64	0.02	54.24	0.05
ACY	0.001	55.24	0.06	25.58	0.03	15.35	0.02	32.06	0.03
ACE	0.001	61.35	0.06	23.27	0.02	28.86	0.03	37.83	0.04
FLU	0.001	63.34	0.06	43.36	0.04	21.41	0.02	42.71	0.04
PHE	0.001	98.59	0.10	69.84	0.07	37.28	0.04	68.57	0.07
ANT	0.010	78.54	0.79	51.99	0.52	35.23	0.35	55.25	0.55
FLT	0.001	223.64	0.22	126.62	0.13	136.75	0.14	162.34	0.16
PYR	0.001	124.67	0.12	93.35	0.09	45.68	0.05	87.90	0.09
BaA	0.100	222.75	22.28	158.84	15.88	82.17	8.22	154.59	15.46
CRY	0.010	272.79	2.73	181.80	1.82	109.08	1.09	187.89	1.88
BbF	0.100	324.89	32.49	232.36	23.24	187.49	18.75	248.24	24.82
BkF	0.100	156.96	15.70	145.89	14.59	95.75	9.58	132.87	13.29
BaP	1.00	236.10	236.10	154.76	154.76	99.89	99.89	163.58	163.58
IND	0.100	191.95	19.20	125.35	12.54	128.24	12.82	148.51	14.85
DBA	1.00	267.54	267.54	170.69	170.69	99.97	99.97	179.40	179.40
BGP	0.010	302.02	3.02	250.00	2.50	193.16	1.93	248.39	2.48
Total carcinogenic activity (TCA)			600.6		397.0		252.9		416.8
Contribution of BaA to the TCA (%)			3.71		4.00		3.25		3.71
Contribution of BbFto the TCA (%)			5.41		5.85		7.41		5.95
Contribution of BkF to the TCA (%)			2.61		3.67		3.79		3.19
Contribution of BaP to the TCA (%)			39.31		38.98		39.50		39.25
Contribution of DBA to the TCA (%)			44.54		42.99		39.52		43.04
Contribution of CRY to the TCA (%)			0.45		0.46		0.43		0.45
Contribution of IND to the TCA (%)			3.20		3.16		5.07		3.56

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
