# Peer review of "Classroom Dust-Bound Polycyclic Aromatic Hydrocarbons in Jeddah Primary Schools, Saudi Arabia: Level, Characteristics and Health Risk Assessment"

_ijerph, 2020, doi:10.3390/ijerph17082779_

Round 1

Reviewer 1 Report

The submitted manuscript discusses the concentration levels, sources and health risks of classroom dust-bound polycyclic aromatic hydrocarbons in Jeddah primary schools, Saudi Arabia. The method of experimentation is suitable for the purpose. It includes a satisfactory and properly selected list of relevant references. It contains interesting data and findings for indoor PAH levels. Overall, the manuscript is well-written, and I suggest that it should be accepted for publication after some minor revisions described below.

Line 55: physicochemical properties

2.1. Sampling sites and collection (lines 111-118): It was mentioned that the details of the study area and its climatic characteristics, sampling sites and the methods of sampling collection and preparation have been reported elsewhere [45]. However, I suppose that a brief description would be necessary about the sampling collection for easier understanding. How many dust samples were collected at the different sampling sites?

Figs. 3-5: Residential schools

2.5.1. Carcinogenic Potency of PAHs (BaPequi) in Classroom Dust: Why the Nisbet and La Goy TEF values were used for the BaPequi calculation?

Author Response

Reply to Reviewer 1:

Comments and Suggestions for Authors

  1. Line 55: physicochemical properties

Response:

Done

  1. Sampling sites and collection (lines 111-118): It was mentioned that the details of the study area and its climatic characteristics, sampling sites and the methods of sampling collection and preparation have been reported elsewhere [45]. However, I suppose that a brief description would be necessary about the sampling collection for easier understanding. How many dust samples were collected at the different sampling sites?

Response:

          Done

  1. 3-5: Residential schools

Response:

Done

  1. Carcinogenic Potency of PAHs (BaPequi) in Classroom Dust: Why the Nisbet and La Goy TEF values were used for the BaPequi calculation?

Response:

It is known that, BaP has potent carcinogenic properties and is considered an enough indicator for the carcinogenicity of the whole PAH compounds. It was also used widely as an air quality indicator, marker for total PAHs exposure and overall carcinogenicity for these PAHs. BaP was selected as a reference chemical, and was assigned a value of one. Based on BaP value, Nisbet and LaGoy (1992) proposed the Toxicity Equivalency Factor (TEF) for each individual PAH compounds. Then, the concentrations of BaPequi in the present study, the amount of carcinogenic potency, for PAH compounds were calculated by multiplying the concentration of the individual measured PAH compounds by its corresponding TEF value.

Reviewer 2 Report

My detailed comments on the manuscript are listed below.

  1. In this study, source apportionment is carried out by diagnostic ratios of individual PAH congeners, which can only be carried out qualitatively but not quantitatively. So, I suggest doing PMF or PCA for further source analysis to obtain source contribution (Rekefu et al. 2018).
  2. Line 223~224: The data expression should be Mean ± SD, and the following expressions are the same.
  3. Line 303: What is the value of IND/IND +BGP? “IND/IND +BGP” change to “IND/(IND +BGP)”, the following statements and Table 2 should be enclosed in parentheses.
  4. Line 429, please check the writing English.

Reference

Rekefu, S., Talifu, D., Gao, B. , Turap, Y., Maihemuti, M., Wang, X., et al. (2018). Polycyclic aromatic hydrocarbons in PM2.5 and PM2.5–10 in Urumqi, China: temporal variations, health risk, and sources. Atmosphere, 9(10).

Author Response

Reply to Reviewer 2

Comments and Suggestions for Authors

  1. In this study, source apportionment is carried out by diagnostic ratios of individual PAH congeners, which can only be carried out qualitatively but not quantitatively. So, I suggest doing PMF or PCA for further source analysis to obtain source contribution (Rekefu et al. 2018).

Response:

Regarding your enquiry about the using of PMF or PCA for further source analysis, quantitatively, to obtain source contribution, in fact the main aim of our submitted paper was to assess the health risks of children exposure for PAHs in classroom dusts using EPA models. When we were planning the sampling sites, we chose schools located in different areas affected mainly by traffic density, which are the main source of PAHs emission. Therefore, source apportionment, qualitatively, was carried out by only using diagnostic ratios of individual PAH congeners in the present study, which was enough to confirm this hypothesis (traffic emission as a main source of PAHs). Moreover, many studies used the diagnostic ratios (in subsection: 3.4. Possible sources of Classroom Dust-Bound PAHs).

  1. Line 223~224: The data expression should be Mean ± SD, and the following expressions are the same.

Response:

Done

  1. Line 303: What is the value of IND/IND +BGP? “IND/IND +BGP” change to “IND/(IND +BGP)”, the following statements and Table 2 should be enclosed in parentheses.

Response:

Done

  1. Line 429, please check the writing English.

Response:

Done

Reviewer 3 Report

In this paper, the authors make use of air conditioning filters in schools to extract PAHs and then to draw conclusions about likely sources, how recent the exposure has been and potential contribution to the overall cancer risk.

The article is generally well written, though some additional proofreading is necessary, particularly of the introductory section, so as to correct numerous typographical and grammatical errors. It also contains a large number of references for a short paper : I'm not sure what the editorial policy is on numbers of references. The study is an application of the same methodology in the same area to a previous paper published in this journal (Shabbaj, I.I., Alghamdi, M.A. and Khoder, M.I. Street Dust—Bound Polycyclic Aromatic Hydrocarbons in a Saudi Coastal City: Status, Profile, Sources, and Human Health Risk Assessment. Int. J. Environ. Res. Public Health 2018, 15, 2397), though here the focus is on dust collected on air conditioning filters, whereas the previous paper focused on PAHs in Street dust. The methodology section refers the reader to the above-mentioned paper for details of the analytical techniques used and the QC/QA parameters. Nevertheless, whilst the same analytical techniques have been used, presumably, a fresh calibration was carried out along with other QC/QA parameters? I think it is important that these are reported for this particular study.

As for the lifetime cancer risk assessment, I'm not convinced that all the assumptions are valid here, particularly the carcinogenic slope factor for inhalation, which is derived from the paper dealing with PAH concentrations in soils. Can the authors make the same assumptions as for the paper that they quote?; it is a very different situation. We are not dealing with resuspended soil particles from contaminated areas: here the calculations have to be made with respect to the concentration of particles in the atmosphere, and deposited on the ground, with the assumption that these particles have the PAH composition characteristics determined by the analysis. Equally, it is really only the fine particulates that will be inhaled and we have no information on what proportion of the dust is comprised of PM 2.5 for example (perhaps, a simple particulate monitor could have been installed in the classroom so that the authors could determine the normal particulate concentrations). The authors will need to justify why their choice of parameters applies to this particular scenario, or to modify the parameters based on more comparable studies.

Is there any reason why error bars are not included on all of the plots?

Author Response

Reply to Reviewer 3

Comments and Suggestions for Authors

  1. The article is generally well written, though some additional proofreading is necessary, particularly of the introductory section, so as to correct numerous typographical and grammatical errors. It also contains a large number of references for a short paper: I'm not sure what the editorial policy is on numbers of references. The study is an application of the same methodology in the same area to a previous paper published in this journal (Shabbaj, I.I., Alghamdi, M.A. and Khoder, M.I. Street Dust—Bound Polycyclic Aromatic Hydrocarbons in a Saudi Coastal City: Status, Profile, Sources, and Human Health Risk Assessment. Int. J. Environ. Res. Public Health 2018, 15, 2397), though here the focus is on dust collected on air conditioning filters, whereas the previous paper focused on PAHs in Street dust. The methodology section refers the reader to the above-mentioned paper for details of the analytical techniques used and the QC/QA parameters. Nevertheless, whilst the same analytical techniques have been used, presumably, a fresh calibration was carried out along with other QC/QA parameters? I think it is important that these are reported for this particular study.

Response:

Done

  1. As for the lifetime cancer risk assessment, I'm not convinced that all the assumptions are valid here, particularly the carcinogenic slope factor for inhalation, which is derived from the paper dealing with PAH concentrations in soils. Can the authors make the same assumptions as for the paper that they quote?; it is a very different situation. We are not dealing with resuspended soil particles from contaminated areas: here the calculations have to be made with respect to the concentration of particles in the atmosphere, and deposited on the ground, with the assumption that these particles have the PAH composition characteristics determined by the analysis. Equally, it is really only the fine particulates that will be inhaled and we have no information on what proportion of the dust is comprised of PM 2.5 for example (perhaps, a simple particulate monitor could have been installed in the classroom so that the authors could determine the normal particulate concentrations). The authors will need to justify why their choice of parameters applies to this particular scenario, or to modify the parameters based on more comparable studies.

Response:

On one hand, indoor air pollutants such as suspended particulate matter (dust), with particle size less than 100 µm, that contains several pollutants like organic and inorganic contamination, can be captured and deposited on the air conditioning filter of school classrooms during air current impact. On the other hand, these deposited particulate matter on the classroom air conditioning filter can be resuspended and also adhere onto indoor surfaces, and then school occupants, including children, can be exposed to them. Therefore, settled dust on the classroom air conditioning filter will represent all of the exposed dust that have migrated to the indoor environment of schools from outdoor environment through doors, ventilation system, windows and AC filters for fresh air, beside the suspended and resuspended of settled dust from indoor particulate sources and children activities. Although the deposited particulate matter on the classroom air conditioning filter are of small size, in the present study we focused and planned to use filter dust sample (size, less than or equal 38-μm) that containing fine particles, that will be inhaled (less than or equal 2.5 -μm (PM2.5)), and also particles from 2.5 -μm up to 38 -μm for  PAHs determination. So, in my opinion, the calculation of the lifetime cancer risk assessment for children exposed to classroom air conditioning filter dust, with particle size less than or equal 38 -μm which containing inhalable fine dust like PM2.5, through all exposure pathways ,inhalation, ingestion and dermal contact, were very important.

  1. Is there any reason why error bars are not included on all of the plots?

Response:

Done, except for figures that containing relative distributions (%) and ILCRs

Round 2

Reviewer 3 Report

I'm not totally convinced that assumptions from a paper on PAHs in soils can be used in the formula to calculate lifetime cancer risks in the present study (the situations are very different, so different assumptions may be needed for use in the equation) . Nevertheless, the results stand on their own, and there is a novel aspect to using dust from air conditioning filter systems to quantify exposure to PAHs.